# Ask Less, See More: Communication-Conditioned Token Pruning for Vehicle-to-Vehicle Cooperative Autonomous Driving with Multimodal Large Language Models

**Shiqi Sun** [1]  **Yantao Lu** [1]  **Bingkun Sun** [2]  **Ning Liu** [3]  **Bo Jiang** [4]  **Ying Zhang** [1]  **Jinchao Chen** [1]  **Chenglie Du** [1]

## Abstract

Multimodal Large Language Models (MLLMs) offer a promising paradigm for vehicle-to-vehicle (V2V) cooperative autonomous driving, enabling language-based decision-making in safety-critical occluded scenarios. However, existing V2V–MLLM frameworks rely on dense token-level sharing and fusion, incurring high communication and inference costs. Moreover, conventional V2V perception methods are limited to feature-sharing paradigms without language reasoning, and existing token pruning strategies fail to consider LiDAR-specific spatial structure and multi-agent fusion. To address these limitations, we propose V2V Communication-Conditioned MLLM Framework (V2V-CCM), a dual-stage cooperative communication framework that broadcasts request messages to all agents and uses them to identify redundant visual tokens. Specifically, Question Semantic Message (QSM) encodes global question intent for question-relevant token selection, while Spatial Coverage Message (SCM) summarizes LiDAR features to identify spatially redundant tokens already observed by other agents. Integrated into dual-stage frameworks, V2V-CCM substantially reduces communication and inference costs while preserving question-relevant tokens and removing spatial redundancy. Extensive experiments on V2V-QA and V2V-GoT-QA demonstrate that V2V-CCM consistently outperforms existing pruning methods and achieves state-of-the-art performance.

[1]School of Computer Science, Northwestern Polytechnical University, Xi'an, China [2]Department of Computer Science, Fudan University, Shanghai, China [3]Beijing Innovation Center of Humanoid Robotics, Beijing, China [4]Didi Chuxing, Beijing, China. Correspondence to: Yantao Lu <yantaolu@nwpu.edu.cn>, Chenglie Du <ducl@nwpu.edu.cn>.

*Proceedings of the 43rd International Conference on Machine Learning*, Seoul, South Korea. PMLR 306, 2026. Copyright 2026 by the author(s).

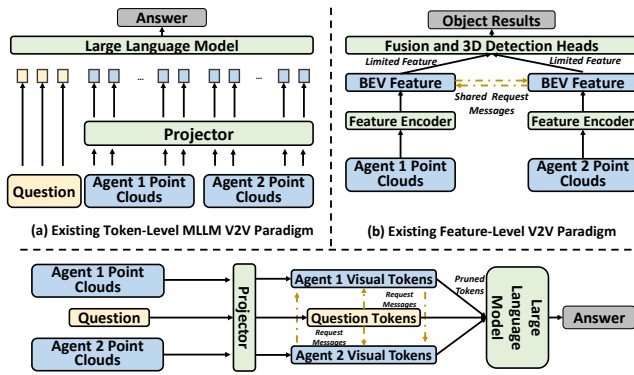

*Figure 1.* **Main differences among V2V paradigms**: (a) existing token-level MLLM-based V2V frameworks rely on dense token sharing for joint reasoning; (b) existing feature-level V2V paradigms exchange intermediate features for collaborative perception; (c) our proposed dual-stage communication-conditioned token pruning paradigm first exchanges compact request messages and then selectively transmits pruned visual tokens for efficient cooperative reasoning.

## 1. Introduction

Multimodal Large Language Models (MLLMs) have emerged as a promising paradigm for autonomous driving by enabling reasoning over sensor inputs and natural language (Sima et al., 2024; Tian et al., 2024b; Xu et al., 2024). However, relying only on single-agent sensor data poses challenges due to severe occlusions and limited local observability. To address this limitation, MLLM-based vehicle-to-vehicle (V2V) cooperative paradigms (Chiu et al., 2026a;b) are proposed, which allows multiple agents to share complementary observations. Unlike conventional feature-level V2V perception frameworks, MLLM-based cooperation fuses LiDAR-specific multimodal tokens from multiple agents. It further adopts language as an interaction interface, enabling vehicles to express driving intents, ask targeted questions about the environment, and coordinate behaviors beyond single-vehicle capabilities. However, MLLM-based token-fusion paradigms incur substantial communication overhead from transmitting tokens and questions, as well as high inference cost from LLM reasoning, as shown in Figure 1. This limitation poses significant

challenges for scalability and real-time deployment.

Although existing V2V paradigms and token pruning techniques offer partial solutions, they cannot fully address the challenges of MLLM-based cooperative driving. Specifically, existing collaborative perception paradigms primarily optimize communication by selecting or compressing shared detection features (Hu et al., 2022; Liu et al., 2020a; Sun et al., 2024). These paradigms are tailored to feature-level sharing and are not well-suited to tokenized, LLM-based reasoning pipelines, particularly when language is used as the communication interface. Meanwhile, existing token pruning methods mainly focus on reducing tokens within standalone LLMs or vision–language models (VLMs) using images or videos. These approaches are designed for single-agent settings and do not consider token pruning in multi-agent, multi-view communication scenarios. More importantly, they overlook the structural properties of LiDAR-derived tokens, such as spatial sparsity and geometric consistency, which are critical for cooperative autonomous driving. As a result, token pruning techniques developed for 2D vision or single-agent scenarios are not well suited for V2V MLLM frameworks that require LiDAR-specific modeling and multi-agent fusion.

To address these challenges, we argue that a *dual-stage communication paradigm*, inspired by conventional V2V perception (Hu et al., 2022; Yang et al., 2023), is necessary. In this paradigm, agents first exchange compact request messages, followed by the selective transmission of requested tokens for fusion, which substantially reduces communication bandwidth. Therefore, we propose V2V Communication-Conditioned MLLM Framework (V2V-CCM), which integrates this dual-stage communication paradigm with token pruning optimization. In this framework, the request message plays a central role by identifying which visual tokens are important for the question and which tokens are redundant and can be omitted. Our key insight is that *"as a highly compressed modality, the natural language–based question serves as a powerful signal for identifying redundant tokens, particularly in communication-constrained scenarios."* Based on this insight, we first design Question Semantic Message (QSM), which compresses the question into a compact token representation and uses it to select question-relevant visual tokens. After identifying question-relevant tokens, we further prune spatially redundant information using Spatial Coverage Message (SCM). This module compresses bird's-eye-view (BEV) features into a structured spatial representation, enabling agents to avoid repeatedly transmitting features from regions that have already been observed. By combining these two messages, our method identifies question-relevant and spatially redundant tokens through Communication-Conditioned Token Pruning (CCTP), enabling effective pruning across all agents. To validate the effectiveness of our proposed framework, extensive experiments are conducted on two existing V2V MLLM-based datasets, including V2V-QA (Chiu et al., 2026a) and V2V-GoT-QA (Chiu et al., 2026b). These datasets encompass a range of V2V cooperative autonomous driving questions, including planning and perception. The main contributions of this work are summarized as follows:

- We propose V2V-CCM, a V2V MLLM framework with dual-stage communication and communication-conditioned token pruning.

- We design two request messages, QSM and SCM, along with communication-conditioned token pruning strategies CCTP.

- We validate the effectiveness of the proposed V2V-CCM on V2V-QA and V2V-GoT-QA across multiple cooperative perception and planning tasks.

## 2. Related Work

### 2.1. LLM-based Autonomous Driving

With the rapid development of LLMs, autonomous driving has increasingly incorporated language as a conditioning modality to enhance reasoning capability across various tasks, including object detection, 3D grounding, question answering, and planning. Early language-based planning methods (Mao et al., 2023a;b) convert driving scenes, detected objects, and ego-vehicle states into textual descriptions, which are then processed by LLMs to generate high-level driving decisions or action suggestions. More recent studies adopt MLLMs (Sima et al., 2024; Xu et al., 2024; Wang et al., 2024; Cui et al., 2025), which encode visual inputs such as images or point clouds into visual tokens and align them with language embeddings for unified vision–language reasoning and question answering.

### 2.2. Collaborative Perception in Autonomous Driving

Early studies on autonomous driving perception mainly rely on LiDAR data from a single ego vehicle and adopt a mono-view perception paradigm. Although these methods achieve strong performance, they often degrade under occlusions and limited fields of view. To address this issue, recent research has increasingly focused on collaborative perception in multi-agent systems, where vehicles share complementary information to improve scene understanding. The release of large-scale datasets such as V2X-SIM (Li et al., 2022), OPV2V (Xu et al., 2022c), DAIR-V2X (Yu et al., 2022), and CoPerception-UAVs (Hu et al., 2022) has enabled systematic development and evaluation of collaborative perception methods (Hu et al., 2022; Xu et al., 2022b;c; Sun et al., 2024; Wang et al., 2020; Yu et al., 2022; Gao et al., 2021; Li et al., 2021; 2023; Liu et al., 2020a;b; Lu et al., 2023; Xu et al., 2022a). These approaches can

be broadly categorized into early-, late-, and intermediate-fusion paradigms, depending on whether raw sensor data, detection outputs, or intermediate features are shared among agents. For instance, DiscoNet (Li et al., 2021) combines early and intermediate fusion through knowledge distillation, while Where2comm (Hu et al., 2022) improves communication efficiency by selectively transmitting spatially important perception features. However, most existing methods rely on feature-level sharing and do not exploit the strong reasoning capability of MLLMs, which operate on token-level representations. More recently, V2V-oriented MLLM frameworks such as V2V-LLM (Chiu et al., 2026a) and V2V-GoT (Chiu et al., 2026b) extend collaborative perception by allowing vehicles to exchange visual tokens and language questions for joint perception, prediction, and planning in challenging scenarios. These token-level sharing approaches demonstrate the potential of token-based, language-driven reasoning for cooperative driving. Nevertheless, they typically adopt dense token sharing and fusion, and do not explicitly address the communication overhead and inference cost introduced by token-level interactions.

### 2.3. Token Pruning

The goal of token pruning is to reduce the number of tokens processed by an MLLM, thereby lowering latency and memory cost while limiting performance degradation. It improves efficiency by preserving informative tokens and removing less informative ones, based on different strategies for estimating token importance, which is especially important for visual tokens that appear in large numbers. For example, DynamicViT (Rao et al., 2021) introduces a trainable prediction module to estimate token importance. Subsequent methods (Fayyaz et al., 2022; Yin et al., 2022) extend this idea by dynamically adjusting the number of retained tokens according to input complexity. To reduce information loss during pruning, some approaches (Huang et al., 2024; Kong et al., 2022) reorganize or collapse pruned tokens into compact representations. However, importance-based methods may assign similar scores to tokens that are close in feature space, increasing the chance that redundant tokens are retained or removed together. An alternative strategy, ToMe (Bolya et al., 2023), prunes tokens by merging highly similar tokens based on feature similarity, but this can introduce spatial distortions that harm fine-grained perception. More recent hybrid methods (Wei et al., 2023; Wu et al., 2023) combine importance scoring with similarity-based merging to address these issues. In contrast, our work focuses on Communication-Conditioned Token Pruning for LiDAR-based, multi-agent V2V MLLM frameworks. By adopting a dual-stage communication strategy that jointly considers question relevance and spatial redundancy of LiDAR-specific tokens, our method enables efficient, communication-conditioned token selection under strict bandwidth constraints.

## 3. Problem Formulation

Current token-level cooperative paradigms increasingly incorporate object tokens and scene tokens from multiple agents to enhance the perception capability required to answer the natural language question $T$. Specifically, given $N$ agents, each agent applies a 3D object detection model to its own LiDAR point cloud $P^1, P^2, \ldots, P^N$ to extract perception features. From each detector, a scene-level feature map $\{\mathcal{F}_s^1, \mathcal{F}_s^2, \ldots, \mathcal{F}_s^N\}$ is obtained, and the corresponding 3D detection results are converted into object-level feature vectors $\{\mathcal{F}_o^1, \mathcal{F}_o^2, \ldots, \mathcal{F}_o^N\}$. To meet the input requirements of MLLMs, these features are further projected into token representations, including scene token sets $\{\mathbf{E}_s^1, \mathbf{E}_s^2, \ldots, \mathbf{E}_s^N\}$ and object token sets $\{\mathbf{E}_o^1, \mathbf{E}_o^2, \ldots, \mathbf{E}_o^N\}$. In addition, a natural language question $T$, which specifies the cooperative task (e.g., perception, prediction, or decision making), is encoded into text tokens $\mathbf{E}_t$ using a text encoder. The resulting token sets, for example $\mathbf{E}_s^k = \{e_s^{k,1}, \ldots, e_s^{k,M}\}$, typically contain a large number of tokens. In practice, visual tokens dominate the total token count compared with language tokens. In a standard V2V MLLM framework, all visual tokens from all agents, together with the text tokens, are concatenated and fed into the MLLM to produce the output $\mathbf{Y}$ (e.g., an answer or a driving suggestion):

$$\mathbf{Y} \sim \mathcal{F}_{\text{MLLM}}\big(\mathbf{E}_t, \mathbf{E}_s^1, \ldots, \mathbf{E}_s^N, \mathbf{E}_o^1, \ldots, \mathbf{E}_o^N\big). \quad (1)$$

Since existing token-level fusion V2V frameworks do not adopt a unified communication strategy, we assume that all tokens, including visual tokens and question tokens, are transmitted during communication. Specifically, each agent shares its LiDAR-specific tokens for cooperative fusion, and the question tokens are shared to specify the cooperative task. Under this assumption, the communication volume for the $i$-th agent can be expressed as

$$\mathcal{B}_i = \sum_{j=1, j\neq i}^{N} \big(\mathbf{E}_s^j + \mathbf{E}_o^j\big) + \mathbf{E}_t. \quad (2)$$

The total number of visual and text tokens, therefore, determines both the communication cost between agents and the inference cost of the MLLM. This motivates the need for effective token pruning strategies and communication-conditioned framework design, as detailed in Section 4.

## 4. Methodology

### 4.1. Overview of V2V-CCM

In this section, we propose *V2V-CCM*, a dual-stage communication framework integrated with *CCTP* for token pruning optimization. In this framework, agents first exchange

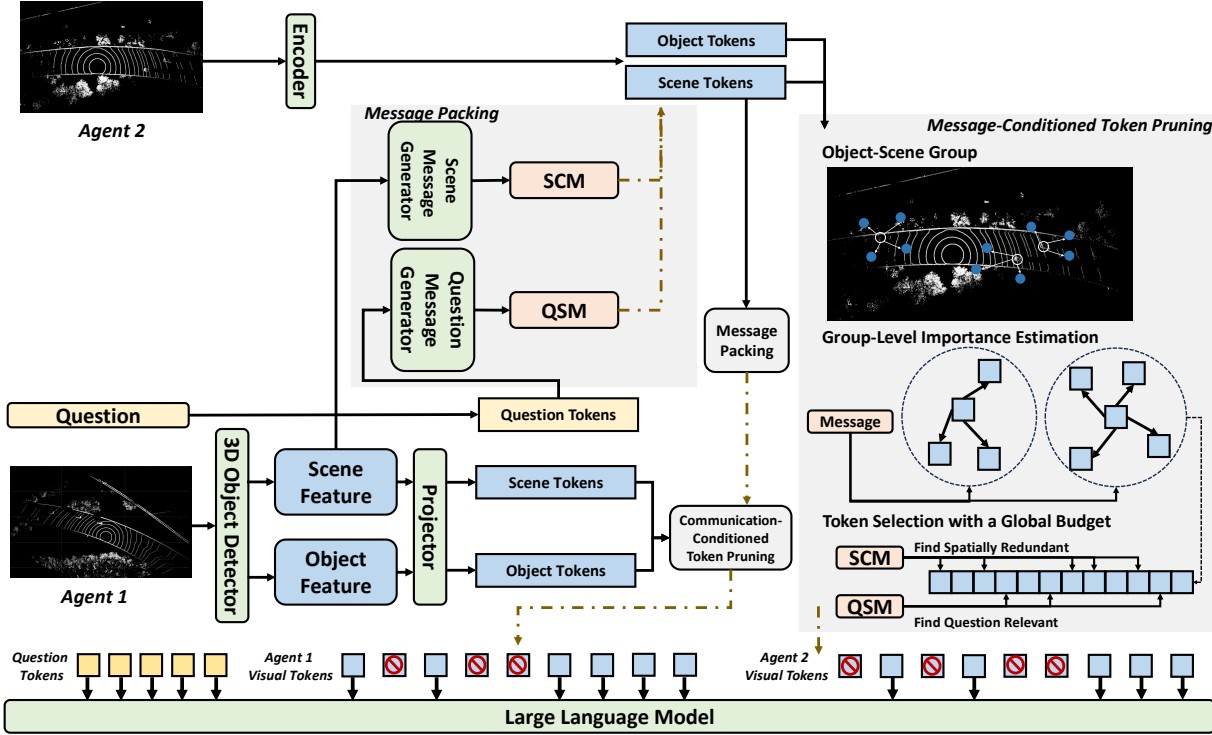

*Figure 2.* **Overview of the proposed V2V-CCM framework**: The framework starts with *Request Message Packing*, where each agent generates compact request messages from its local observations and the input question, including the Question Semantic Message (QSM) and Spatial Coverage Message (SCM). These request messages are then exchanged among agents through communication and used to guide *Communication-Conditioned Token Pruning*. Specifically, token pruning is performed in three stages: *Object–Scene Grouping*, where object tokens and surrounding scene tokens are grouped into spatial regions; *Group-Level Importance Estimation*, where region importance is evaluated based on the received request messages; and *Token Selection with a Global Budget*, where question-relevant tokens are retained and spatially redundant tokens are removed under a global token budget constraint. The remaining visual tokens, together with the question tokens, are finally fed into the large language model to generate the output. In the figure, *black solid lines* indicate model computation flows, while *yellow dashed lines* represent inter-agent communication of request messages.

a compact request message and then selectively transmit question-relevant tokens for fusion, which substantially reduces communication bandwidth. As illustrated in Figure 2, the overall framework consists of two main components. First, the communication stage exchanges request messages that guide each agent on what tokens should be pruned, as described in Section 4.2. Second, the pruning stage, implemented by CCTP, determines how visual tokens are pruned based on the received messages, as detailed in Section 4.3.

### 4.2. Request Message Packing for Token Pruning

Due to bandwidth constraints, the communication module should transmit only compact request messages that carry the most informative content. Such messages not only reduce communication overhead but also enable other agents to identify which information should be preserved and which parts are redundant for cooperative reasoning. To address this issue, we define two key request messages for communication-efficient cooperative reasoning. 1) **Question Semantic Message** (QSM) is derived from the input

question and compresses it into a compact representation of the task intent. It captures the core intent of the cooperative question and guides the selection of question-relevant visual tokens. 2) **Spatial Coverage Message** (SCM) is derived from LiDAR point clouds and tokenizes BEV features into a spatial representation. This message identifies spatial regions that are already observed, allowing agents to avoid repeatedly transmitting spatially redundant visual tokens.

*QSM* is represented as a single token derived from the question tokens and encodes the global semantics of the question. Specifically, directly using the full question context as a request message would introduce unnecessary communication overhead. It may also lead to inefficient attention allocation, since less important question terms can dominate the selection process. To address this issue, we construct a compact representation that captures the global semantics of the question. Specifically, for the $i$-th agent, we apply a generator to the question token embeddings to produce a compact semantic vector:

$$\mathcal{M}_t^i = \Phi_{\text{generator}}^t(\mathbf{E}_t), \qquad (3)$$

where $\mathcal{M}_t^i$ denotes the QSM generated by the $i$-th agent, and $\Phi_{\text{generator}}^t(\cdot)$ represents a compression token projector, e.g., mean pooling. It encodes the overall question intent and serves as a reliable request message for selecting question-relevant visual tokens.

*SCM* represents the spatial regions that are already perceptually known by an agent. Intuitively, in cooperative driving, a key goal is to complement regions that are occluded or outside the limited field of view, in order to recover objects that may be missed by a single agent. Accordingly, transmitted messages should guide other agents to prioritize tokens from unknown or occluded regions, rather than from areas that are already well observed by the ego agent. If an object or region has been reliably perceived by one agent, it does not need to be redundantly shared by others. Based on this principle, SCM tokenizes the LiDAR BEV map into a compact spatial token that explicitly encodes which regions and objects are already known. This design enables cooperative agents to selectively transmit only missing or complementary spatial information, instead of repeatedly sharing redundant observations. Similarly to the QSM, given the scene token set $\mathbf{E}_s^i = \{\mathbf{E}_s^1, \mathbf{E}_s^2, \ldots, \mathbf{E}_s^N\}$ of the $i$-th agent, the corresponding spatial message is computed as

$$\mathcal{M}_s^i = \Phi_{\text{generator}}^s(\mathbf{E}_s^i), \qquad (4)$$

where $\mathcal{M}_s^i$ denotes the SCM generated by the $i$-th agent, and $\Phi_{\text{generator}}^s(\cdot)$ is a compression token projector, such as mean pooling. This module compresses the scene token set into a single token that represents the BEV spatial map of the $i$-th agent. Therefore, the packed request message consists of two complementary components: the QSM and the SCM, which together guide the following token pruning.

## 4.3. Communication-Conditioned Token Pruning

In this section, we introduce Communication-Conditioned Token Pruning (CCTP), which performs token pruning by jointly considering question relevance and spatial redundancy. After aggregating the communication messages, including QSM $\mathcal{M}_t$ and SCM $\mathcal{M}_s$, CCTP selects visual tokens that are highly relevant to $\mathcal{M}_t$ while suppressing tokens from spatial regions already covered by $\mathcal{M}_s$. Since LiDAR-based representations include both object-level and scene-level tokens, directly pruning tokens at the individual level may introduce bias toward either objects or background regions. To address this issue, we group each object token with its surrounding scene region, enabling region-level pruning guided by the communicated messages to identify redundant tokens. Specifically, CCTP consists of three stages.

First, *object–scene grouping* is applied, enabling token importance to be analyzed at the region level rather than at the individual token level. Second, *group-level importance*

*estimation* is performed. For each group, we apply mean pooling to obtain a group representation and compute both a question-relevance score and a spatial-redundancy score, which together reflect the importance of the tokens at the region level. Third, *token selection with a global budget* is conducted. Tokens are scored by maximizing their relevance to QSM $\mathcal{M}_t$ and minimizing their relevance to SCM $\mathcal{M}_s$. At the same time, we ensure that the importance of these tokens is proportional to their region-level importance. Details of each stage are described below. Since strategy is the same for all agents, we omit the agent index $i$ for clarity.

*Object–Scene Grouping.* We first construct token groups based on the LiDAR point cloud. Each object token is treated as a group center. Scene tokens located within a spatial neighborhood of the object, defined by their LiDAR coordinates, are assigned to the same group. Formally, let $\mathcal{G} = \{g^1, g^2, \ldots, g^M\}$ denote the set of groups. For each group $g^m \in \mathcal{G}$, we denote the set of object tokens as $g_o^m$ and the set of scene tokens as $g_s^m$. The group $g^m$ is defined as

$$g^m = g_o^m \cup g_s^m.$$

These groups jointly cover all tokens, such that

$$\bigcup_{m=1}^{M} g_o^m = \mathbf{E}_o, \quad \bigcup_{m=1}^{M} g_s^m = \mathbf{E}_s, \qquad (5)$$

where $\mathbf{E}_o$ and $\mathbf{E}_s$ denote the full sets of object tokens and scene tokens, respectively.

*Group-level Importance Estimation.* We next introduce how to estimate the importance of each group. For each group $g^m$, we first aggregate all tokens within the group using mean pooling to obtain a group-level representation:

$$\bar{g}^m = \frac{1}{|g^m|} \sum_{e \in g^m} e, \qquad (6)$$

where $|g^m|$ denotes the number of tokens in group $g^m$. Based on the group representation $\bar{g}^m$, we compute two importance scores. The question-relevance score $\mathcal{I}_t^m$ measures how relevant the group is to the question and is defined as the cosine similarity between $\bar{g}_m$ and the QSM $\mathcal{M}_t$:

$$\mathcal{I}_t^m = \frac{\mathcal{M}_t \cdot \bar{g}^m}{\|\mathcal{M}_t\|_2 \, \|\bar{g}^m\|_2}, \quad \forall m \in \{1, 2, \ldots, M\}. \quad (7)$$

Similarly, the spatial-redundancy score $\mathcal{I}_s^m$ measures how redundant the group is with respect to the known spatial regions encoded in SCM $\mathcal{M}_s$:

$$\mathcal{I}_s^m = \frac{\mathcal{M}_s \cdot \bar{g}^m}{\|\mathcal{M}_s\|_2 \, \|\bar{g}^m\|_2}, \quad \forall m \in \{1, 2, \ldots, M\}. \quad (8)$$

Finally, the group-level importance score is defined by jointly considering question relevance and spatial redundancy:

$$\mathcal{I}^m = \mathcal{I}_t^m - \gamma \mathcal{I}_s^m, \qquad (9)$$

where $\gamma$ is a weighting factor that balances task relevance and spatial redundancy. Groups with higher $\mathcal{I}^m$ are assigned higher importance and retain more tokens.

*Token Selection with a Global Budget.* After estimating group-level importance, we perform token selection within each group. For each visual token $e \in g^m$, we compute two token-level scores, following the definitions in Eq. 9 and Eq. 8. The question-relevance score $\mathcal{S}_t^i$ is defined as

$$\mathcal{S}_t^i = \frac{\mathcal{M}_t \cdot e}{\|\mathcal{M}_t\|_2 \|e\|_2}, \quad \forall i \in \{1, 2, \ldots, |g^m|\}. \quad (10)$$

Similarly, the spatial-redundancy score $\mathcal{S}_s^i$ is defined as

$$\mathcal{S}_s^i = \frac{\mathcal{M}_s \cdot e}{\|\mathcal{M}_s\|_2 \|e\|_2}, \quad \forall i \in \{1, 2, \ldots, |g^m|\}. \quad (11)$$

Token selection is then formulated as a constrained optimization problem. Given a total budget of $K$ retained tokens, we select token subsets $\tilde{g}^m \subseteq g^m$ for all groups $m = 1, \ldots, M$. For each token $e \in g^m$, we define its token importance scores as

$$\mathcal{S}_e = \mathcal{S}_t^i - \lambda \mathcal{S}_s^i, \quad (12)$$

where $\lambda$ controls the trade-off between task relevance and spatial redundancy. We then solve the following budgeted selection problem:

$$\max_{\{\tilde{g}^m\}_{m=1}^M} \sum_{m=1}^M \mathcal{I}^m \sum_{e \in \tilde{g}^m} \mathcal{S}_e \quad (13)$$

$$\text{s.t.} \quad \sum_{m=1}^M |\tilde{g}^m| = K \leq \mathcal{B}, \quad (14)$$

$$\tilde{g}^m \subseteq g^m, \quad m = 1, \ldots, M. \quad (15)$$

This formulation promotes the selection of question-relevant tokens while suppressing spatially redundant tokens, and allocates more tokens to groups with higher importance scores $\mathcal{I}^m$. The resulting token selection can be efficiently solved using a top-$K$ ranking strategy.

# 5. Experimental Results

## 5.1. Baseline Models and Pruning Methods

*Baseline Models.* We evaluate our pruning method using V2V-LLM (Chiu et al., 2026a) and V2V-GoT (Chiu et al., 2026b) as baseline models. Both are MLLM-based V2V cooperative driving frameworks and share a similar overall architecture. Specifically, both baselines adopt LLaVA (Liu et al., 2023) as the core MLLM and use a LiDAR-based 3D object detector, PointPillars (Lang et al., 2019), to extract perception features from point clouds. The extracted visual features are projected into the language embedding space using an MLP projector. These visual tokens are combined with the language tokens from the input question and

fed into the LLM, which aggregates information from all connected agents to generate the final output. V2V-GoT further extends V2V-LLM by incorporating perception features from both the current and previous timesteps of all agents when generating visual tokens.

*Baseline Pruning Methods.* To comprehensively evaluate our method, we compare it with several representative token pruning baselines, including Random (Yao et al., 2022), Token Merging (ToMe) (Bolya et al., 2023), VisionZip (Yang et al., 2025), and DivPruner (Alvar et al., 2025). The Random baseline serves as a lower-bound reference, where tokens are removed uniformly at random according to the target pruning ratio. ToMe (Bolya et al., 2023) is a lightweight method that iteratively merges visually similar tokens using an efficient matching strategy. VisionZip (Yang et al., 2025) compresses tokens by identifying semantically redundant visual tokens based on attention patterns within transformer layers, making it a strong baseline for vision-centric pruning. DivPruner (Alvar et al., 2025) prunes tokens through diversity-aware selection to preserve informative and diverse representations. Since these pruning methods were originally designed for ViT-based vision models and rely on attention scores, we adapt their algorithms to operate in our LiDAR-based setting. All baseline methods are evaluated under the same token pruning ratios and compared directly with our proposed V2V-CCM.

## 5.2. Datasets and Evaluation Metrics

We conduct our experiments on the V2V-QA (Chiu et al., 2026a) and V2V-GoT-QA (Chiu et al., 2026b) datasets, which are large-scale real-world benchmarks for cooperative autonomous driving with multi-agent LiDAR.

*V2V-QA.* V2V-QA is built on the V2V4Real (Xu et al., 2023) and V2X-Real (Xiang et al., 2024) datasets and provides question–answer pairs under two scenario splits: V2V-split and V2X-split. It covers grounding tasks (Q1–Q3), notable object identification (Q4), and planning (Q5). In total, V2V-QA contains approximately 1.45M QA pairs, with over 7,105 training and 1,993 testing frames from V2V4Real, and 5,772 training and 1,253 testing frames from V2X-Real. Each frame includes around 30 queries, enabling comprehensive evaluation of cooperative perception and planning. Due to dataset availability constraints, we report results only on the V2V-split.

*V2V-GoT-QA.* To evaluate multi-step reasoning, we further conduct experiments on the V2V-GoT-QA dataset. V2V-GoT-QA extends the V2V-LLM setting by introducing a graph-of-thoughts style QA structure. It is also derived from V2V4Real, but augments each frame with nine types of questions spanning perception, prediction, and planning (Q1–Q9). The dataset contains 110,610 training and 31,014 testing QA pairs. Answers from earlier perception and pre-

*Table 1.* Comparison of V2V-CCM with baseline pruning methods under different pruning ratios (0%, 30%, 50%, 70%) on V2V-QA and V2V-GoT-QA datasets (V2V-split). For V2V-QA, we evaluate grounding tasks (Q1–Q3 with average $Q_{Gr}$), notable object identification (Q4), and planning (Q5 with L2 distance error and collision rate) with V2V-LLM as baseline models. For V2V-GoT-QA, we report the final planning task Q9 (L2 distance error and collision rate) with V2V-GoT as baseline models. The base model block shows results without pruning; subsequent blocks show 30%, 50%, and 70% pruning with baseline pruning methods (Random (Yao et al., 2022), ToMe (Bolya et al., 2023), VisionZip (Yang et al., 2025), DivPruner (Alvar et al., 2025)) and our proposed method. "-" indicates this model is not implemented in this dataset.

| Method | V2V-QA V2V-split | | | | | | | V2V-GoT-QA V2V-split | |
| | Q1 | Q2 | Q3 | $Q_{Gr}$ | Q4 | Q5 | | Q9 | |
| | F1 ↑ | F1 ↑ | F1 ↑ | F1 ↑ | F1 ↑ | L2 (m) ↓ | CR (%) ↓ | L2 (m) ↓ | CR (%) ↓ |
|---|---|---|---|---|---|---|---|---|---|
| *Base model (no pruning)* | | | | | | | | | |
| V2V-LLM | 70.0 | 30.8 | 21.2 | 40.7 | 59.7 | 4.99 | 3.00 | - | - |
| V2V-GoT | - | - | - | - | - | - | - | 2.62 | 1.83 |
| *30% pruning* | | | | | | | | | |
| Random (Yao et al., 2022) | 66.4 | 26.3 | 17.2 | 36.6 | 51.2 | 6.78 | 4.98 | 4.79 | 3.71 |
| ToMe (Bolya et al., 2023) | 68.9 | 28.4 | 19.3 | 38.9 | 55.0 | 5.54 | 4.55 | 4.13 | 3.18 |
| VisionZip (Yang et al., 2025) | 69.6 | 30.2 | 21.1 | 40.3 | 58.3 | 5.16 | 3.44 | 3.15 | 2.43 |
| DivPruner (Alvar et al., 2025) | 69.9 | 30.5 | 19.7 | 40.0 | 59.1 | 5.25 | 3.51 | 3.37 | 2.59 |
| V2V-CCM (Ours) | 69.8 | 30.7 | 21.0 | 40.5 | 59.3 | 5.05 | 3.17 | 2.65 | 1.84 |
| *50% pruning* | | | | | | | | | |
| Random (Yao et al., 2022) | 64.1 | 25.1 | 16.3 | 35.2 | 48.0 | 7.50 | 6.50 | 6.93 | 6.31 |
| ToMe (Bolya et al., 2023) | 66.4 | 26.9 | 18.5 | 37.3 | 53.2 | 5.91 | 5.17 | 5.42 | 4.84 |
| VisionZip (Yang et al., 2025) | 68.8 | 29.1 | 19.4 | 39.1 | 56.8 | 5.72 | 5.88 | 4.24 | 3.68 |
| DivPruner (Alvar et al., 2025) | 68.5 | 28.9 | 20.0 | 39.1 | 57.3 | 5.87 | 5.79 | 4.51 | 3.94 |
| V2V-CCM (Ours) | 69.5 | 29.9 | 20.4 | 39.9 | 57.6 | 5.17 | 5.01 | 2.88 | 2.31 |
| *70% pruning* | | | | | | | | | |
| Random (Yao et al., 2022) | 59.8 | 22.2 | 14.3 | 32.1 | 45.4 | 9.94 | 10.68 | 9.21 | 9.84 |
| ToMe (Bolya et al., 2023) | 63.0 | 24.6 | 15.4 | 34.3 | 49.3 | 7.58 | 7.80 | 7.11 | 7.52 |
| VisionZip (Yang et al., 2025) | 67.5 | 27.8 | 17.2 | 37.5 | 53.8 | 6.97 | 8.15 | 5.48 | 6.09 |
| DivPruner (Alvar et al., 2025) | 66.8 | 27.2 | 16.8 | 36.9 | 52.9 | 7.35 | 7.68 | 5.83 | 6.42 |
| V2V-CCM (Ours) | 69.1 | 28.3 | 18.8 | 38.7 | 54.4 | 5.21 | 6.28 | 3.01 | 2.73 |

diction queries are reused as context for subsequent planning questions, forming a challenging testbed for long-horizon cooperative reasoning under token pruning.

*Metrics for V2V-QA.* Following prior work (Tian et al., 2024a; Wang et al., 2024) and the settings of V2V-QA, we evaluate grounding questions (Q1–Q3) and the notable object identification question (Q4) using precision, recall, and F1 score. Both ground-truth answers and model outputs consist of object center locations. A prediction is considered a true positive if the Euclidean distance between the predicted center and the ground-truth center is less than 4 meters, which corresponds to a typical vehicle length. For the planning task (Q5), we report the average L2 error and collision rate (CR). Each prediction consists of six future waypoints, and the L2 error is computed over these waypoints. For collision evaluation, we assume each agent has a bounding box of 4 m (length), 2 m (width), and 1.5 m (height). A collision is counted if the Intersection-over-Union (IoU) between the predicted bounding box and any ground-truth object bounding box is greater than zero.

*Metrics for V2V-GoT-QA.* Following prior studies (Chiu et al., 2026a;b; Sima et al., 2024) and the settings of V2V-GoT-QA, we adopt task-specific evaluation metrics. For

grounding and object identification tasks (Q1–Q4), we report the F1 score. For Q6, we report binary classification accuracy, evaluating whether the model correctly identifies other agents as notable objects. For planning-related tasks, we report the L2 error for Q5 and Q7, the L1 error for Q8, and both L2 error and CR for Q9.

## 5.3. Main Results on V2V-QA and V2V-GoT-QA

The main experimental results are reported in Table 1, where we compare V2V-CCM with baseline token pruning methods under different pruning ratios (30%, 50%, and 70%) on the V2V-QA and V2V-GoT-QA benchmarks.

*Results on V2V-QA (V2V-split).* As shown in Table 1, general pruning strategies that are not tailored to cooperative V2V reasoning lead to clear performance degradation across both perception and planning tasks. In contrast, V2V-CCM consistently achieves the best overall trade-off across perception and planning tasks under all pruning settings. At 30% and 50% pruning, V2V-CCM maintains performance close to the unpruned baseline on grounding and object identification tasks, while achieving the lowest planning error (5.05 m) and CR (3.17%) at the 30% pruning ratio

*Table 2.* Efficiency evaluation under different pruning ratios. Communication bandwidth, computation cost, and TTFT are reduced consistently as the pruning ratio increases. Speedup is computed based on TTFT.

| Model | Pruning Ratio | Comm. BW (MB) ↓ | TFLOPs ↓ | TTFT (ms) ↓ | Speedup ↑ |
|---|---|---|---|---|---|
| V2V-LLM | 0% | 0.4068 | 15.12 | 45.5 | 1.00× |
|  | 30% | 0.2906 | 10.60 | 36.0 | 1.26× |
|  | 50% | 0.2037 | 7.99 | 30.2 | 1.51× |
|  | 70% | 0.1251 | 4.50 | 22.0 | 2.07× |
| V2V-GoT | 0% | 0.4068 | 16.04 | 48.3 | 1.00× |
|  | 30% | 0.2913 | 11.20 | 38.0 | 1.27× |
|  | 50% | 0.2056 | 8.90 | 31.9 | 1.51× |
|  | 70% | 0.1278 | 5.00 | 23.0 | 2.10× |

*Table 3.* Ablation study on the effect of hyperparameter $\lambda$ in V2V-CCM on V2V-QA V2V-split dataset under 70% pruning ratio. $\lambda$ controls the trade-off between question relevance guided by (QSM) and spatially redundant guided by (SCM) in token selection. We report the planning task (Q5) performance measured by L2 distance error (m) and collision rate (%). The first row shows the baseline performance without pruning.

| $\lambda$ | Q5 (Planning) | |
|---|---|---|
|  | L2 (m) ↓ | CR (%) ↓ |
| *No pruning* | 4.99 | 3.00 |
| 0.0 | 6.08 | 7.54 |
| 0.5 | 5.61 | 6.81 |
| 0.8 | 5.46 | 6.53 |
| 1.0 | 5.43 | 6.40 |
| 1.2 | 5.21 | 6.28 |
| 1.5 | 5.34 | 6.32 |

among all pruning methods. Even under aggressive pruning at 70%, V2V-CCM significantly outperforms all baseline methods, demonstrating strong robustness to token reduction. These results indicate that V2V-CCM effectively preserves question-relevant information while removing spatially redundant tokens, even when a large fraction of visual tokens is pruned.

*Results on V2V-GoT-QA.* We further evaluate our method on the V2V-GoT-QA benchmark, which emphasizes cooperative reasoning with the graph of thoughts. As reported in Table 1, baseline pruning methods suffer from severe performance degradation on the final planning task (Q9), with sharply increased L2 error and CR as pruning becomes more aggressive. In contrast, V2V-CCM consistently achieves the best performance across all pruning ratios. At 30% pruning, V2V-CCM nearly matches the unpruned baseline, achieving an L2 error of 2.65 m and a CR of 1.84%. At 50% and 70% pruning, V2V-CCM substantially outperforms all baselines, maintaining much lower planning error and CR than Random, ToMe, VisionZip, and DivPruner.

These results demonstrate that V2V-CCM generalizes well across different datasets and LLM-based cooperative driving settings, including both V2V-QA and V2V-GoT-QA. Across both datasets and all pruning ratios, V2V-CCM consistently achieves a better balance between efficiency and performance than existing token pruning methods.

### 5.4. Efficiency Evaluation

We further evaluate the efficiency of the proposed token pruning strategy in terms of communication bandwidth, computational cost, and inference latency. Table 2 reports representative results for V2V-LLM and V2V-GoT under different pruning ratios. In the original V2V-LLM and V2V-GoT protocols, communication bandwidth is measured by the size of transmitted visual features rather than the number of tokens. For a fair and consistent comparison, we follow the same protocol and compute bandwidth based on

the dimensionality of the transmitted features. Although our method prunes visual tokens, each token is projected from a corresponding visual feature representation. Pruned tokens can therefore be mapped back to their related features, which are omitted from transmission. Under this bandwidth calculation protocol, token pruning directly reduces the transmitted feature volume. As the pruning ratio increases, both V2V-LLM and V2V-GoT consistently achieve lower communication bandwidth, fewer TFLOPs, and shorter time to first token (TTFT). At a pruning ratio of 70%, the proposed strategy reduces communication bandwidth by approximately 70% and yields more than a 2× TTFT speedup for both models, while preserving strong task performance.

We further evaluate TTFT on both server-grade and edge devices. As shown in Table 2, on an H200 GPU, 70% pruning reduces the TTFT of V2V-LLM from 45.5 ms to 22.0 ms, and that of V2V-GoT from 48.3 ms to 23.0 ms. On a Jetson AGX Orin, the end-to-end latency of V2V-LLM decreases from 5.33 s to 4.83 s, while that of V2V-GoT decreases from 6.14 s to 5.55 s. Although the dual-stage communication design introduces an additional request-message exchange, the reduction in transmitted visual tokens and LLM input length lowers the overall computational burden and latency on both server-grade and edge devices. These results demonstrate the practical efficiency of the proposed method for real-world V2V cooperative driving.

### 5.5. Ablation Study

We conduct an ablation study to analyze the effect of the balance parameter $\lambda$ in Eq. 12. The parameter $\lambda$ controls the trade-off between question-relevance scores and spatial redundancy scores in CCTP. A smaller $\lambda$ emphasizes question relevance to the question, while a larger $\lambda$ places more weight on suppressing spatially redundant tokens.

The results are reported in Table 3, evaluated on the V2V-QA V2V-split under a fixed pruning ratio of 70%. We focus on the planning task (Q5) and report the L2 distance error (m) and collision rate (CR, %). The first row shows the baseline performance without token pruning. As shown in Table 3, applying token pruning with an appropriate $\lambda$ substantially mitigates the degradation caused by aggressive pruning. As $\lambda$ increases, both L2 error and collision rate steadily decrease, reaching the best performance at moderate values (around $\lambda = 1.0$–$1.2$). Further increasing $\lambda$ does not lead to additional improvements and may slightly degrade performance, suggesting that excessive emphasis on spatial redundancy can remove question-relevant tokens.

## 6. Discussion and Conclusion

V2V cooperative reasoning with multimodal large language models has emerged as a powerful paradigm for autonomous driving, enabling language-driven joint perception, prediction, and planning. However, existing V2V-LLM frameworks typically rely on dense token sharing among agents, which introduces substantial communication overhead and limits scalability under realistic bandwidth constraints. To address this challenge, we propose a dual-stage communication framework, V2V-CCM, with message-conditioned token pruning CCTP. The key idea is to enable agents to first exchange compact request messages and then selectively transmit only the requested tokens for fusion, which significantly reduces communication bandwidth. Within this framework, we design two complementary request messages: QSM that captures question intent, and SCM that encodes already observed regions. These messages jointly guide token pruning by preserving question-relevant information while removing spatially redundant tokens. Extensive experiments on V2V-QA and V2V-GoT-QA demonstrate that the proposed method consistently outperforms existing token pruning baselines across different pruning ratios. Our approach significantly reduces communication cost while maintaining strong performance on both perception and long-horizon planning tasks. These results validate the effectiveness of message-conditioned token pruning for scalable V2V-LLM-based cooperative driving.

While the proposed framework effectively reduces communication overhead, the inference cost of large language models remains high, especially as the number of agents and task complexity increase. In this work, we focus on pruning tokens at the communication stage, but do not explicitly optimize token usage within the LLM inference process itself. Future work will explore inference-stage token pruning and adaptive reasoning strategies, aiming to jointly reduce communication and computation costs. We believe that integrating communication-aware and inference-aware optimization will be a key step toward practical, large-scale

deployment of V2V-LLM systems.

## Impact Statement

This paper presents research aimed at improving both communication and inference efficiency in large language model–based V2V cooperative autonomous driving systems. By reducing communication overhead through message-conditioned token pruning, the proposed method facilitates more practical deployment of cooperative perception and planning under realistic bandwidth constraints.

The potential positive societal impacts of this work include enhanced safety, robustness, and efficiency of V2V autonomous driving systems, particularly in complex traffic environments where inter-vehicle cooperation is critical. More efficient communication may also help reduce energy consumption and infrastructure requirements in large-scale intelligent transportation systems. This work does not introduce new data collection processes, does not involve personal or sensitive data, and does not alter decision-making mechanisms beyond existing autonomous driving frameworks.

Overall, we believe the societal implications of this work are aligned with the established goals of advancing machine learning and autonomous systems, and we do not foresee ethical concerns beyond those commonly associated with research in this domain.

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
