# OpenReview forum: "Ask Less, See More: Communication-Conditioned Token Pruning for Vehicle-to-Vehicle Cooperative Autonomous Driving with Multimodal Large Language Models"
_ICML.cc/2026/Conference — ICML 2026 regular_

### Official Review · Reviewer_BZpB · 2026-02-14

**Soundness:** 3
**Presentation:** 1
**Significance:** 2
**Originality:** 3
**Overall Recommendation:** 4
**Confidence:** 4

**Summary:**

In this paper, the authors propose V2V-CCM, an MLLM-based framework for communication-efficient cooperative reasoning tasks. The framework introduces message-guided token pruning to address communication cost challenges in V2V scenarios. Specifically, two key request messages, QSM and SCM, are defined, and a Message-Conditioned Token Pruning (MCTP) strategy is proposed. Experimental results on the V2V-QA and V2V-GoT-QA datasets demonstrate the effectiveness and performance advantages of the proposed approach.

**Compliance With Llm Reviewing Policy:**

Affirmed.

**Final Justification:**

The experimental results address my concerns. I have carefully read the authors' response as well as the discussion with other reviewers. I have decided to raise my score to relatively positive.

**Key Questions For Authors:**

1. How do the communication cost and inference latency compare before and after token pruning?
2. How can the proposed MCTP method be extended to multi-agent scenarios involving more than two vehicles?
3. What is the performance impact when token importance is analyzed at the instance level? It would be helpful to include an ablation study on object-scene grouping and group-level importance estimation.

**Limitations:**

yes

**Strengths And Weaknesses:**

Strength
1. The V2V-CCM framework targets communication-efficient cooperative reasoning, which is essential for practical V2V applications.
2. Two key request messages are clearly defined: QSM captures question intent, while SCM encodes observed regions.
3. The proposed MCTP strategy considers both question relevance and spatial redundancy when pruning tokens.
4. V2V-CCM outperforms baseline methods on two cooperative reasoning benchmarks.


Weakness
1. Tokens from observed regions are treated as redundant; however, features of the same object from different viewpoints can still be valuable in cooperative perception tasks. Conventional methods typically consider multi-view feature fusion for both co-aware object enhancement and unaware object complementation [1].

     [1] QUEST: Query Stream for Practical Cooperative Perception. ICRA, 2024.
2. The proposed approach introduces an additional communication stage, which may increase system complexity and reduce robustness in practical deployments.
3. There are several minor issues in the current version: (1) QSM in line 036 should be SCM. (2) An unknown reference appears in line 107. (3) “Find spatially redunat” in Fig. 2 should be corrected.
4. Multiple terms are used interchangeably for the same concept (e.g., message-guided, communication-conditioned, and message-conditioned), which may cause confusion.

---

> ### Author Rebuttal · Authors · 2026-03-30
>
> We appreciate the professional and insightful comments, and address each point as follows:
>
> **W1 Multi-view preservation**
>
> We thank the reviewer for this suggestion. We agree that adding a discussion on limitations would further improve the paper. Since our method operates on scene-level features, tokens observed from different viewpoints tend to have lower similarity than those from the same viewpoint. As a result, cross-view tokens are less likely to be pruned and are more likely to be preserved, which supports multi-view feature fusion. In addition, QSM helps ensure that tokens relevant to the query are retained, even when they correspond to the same scene from different viewpoints. Overall, tokens that are irrelevant to the query or come from similar viewpoints are pruned first, while complementary multi-view tokens are preserved as much as possible.
>
> ---
>
> **W2 Latency comparison**
>
> We thank the reviewer for this observation. We agree that there is a trade-off between a dual-stage paradigm and full-token sharing. However, the dual-stage design reduces communication bandwidth and redundancy. As discussed in Q2, full-token broadcasting leads to quadratic communication cost as the number of agents increases. Moreover, directly sharing tokens without prior coordination may introduce redundancy and inefficiency. In contrast, the “request-then-share” paradigm enables selective communication and is more scalable. As further analyzed in W3 & Q3 of Review Q3ns, although it introduces additional communication overhead, this is largely offset by reduced inference time from token pruning, resulting in lower bandwidth and stable overall latency.
>
> ---
>
> **W3 & W4 Writing issues**
>
> We thank the reviewer for these comments and have corrected the typographical errors and inconsistent terminology, which will be reflected in the camera-ready version.
>
> ---
>
> **Q1 Efficiency metrics**
>
> We agree that reporting efficiency metrics is important. We provide representative results showing the impact of token pruning on communication and computation:
>
> | Model    | Pruning Ratio | Comm. BW (MB) ↓ | TFLOPs ↓ | TTFT (ms) ↓ | Speedup ↑ |
> |----------|---------------|-----------------|----------|-------------|-----------|
> | V2V-LLM  | No Pruning    | 0.4068          | 15.12   | 45.5        | 1.00×     |
> |          | 30%           | 0.285           | 10.6    | 36.0        | 1.26×     |
> |          | 50%           | 0.2034          | 7.99    | 30.2        | 1.51×     |
> |          | 70%           | 0.122           | 4.5     | 22.0        | 2.07×     |
> | V2V-GoT  | No Pruning    | 0.4068          | 16.04   | 48.3        | 1.00×     |
> |          | 30%           | 0.285           | 11.2   | 38.0        | 1.27×     |
> |          | 50%           | 0.2034          | 8.90    | 31.9        | 1.51×     |
> |          | 70%           | 0.122           | 5.0     | 23.0        | 2.10×     |
>
> These results show that higher pruning ratios consistently reduce communication cost, computation, and latency.
>
> ---
>
> **Q2 Multi-agent scalability**
>
> Since the available datasets only include scenarios with two agents, our experiments are limited to the two-agent setting. However, our framework is designed to extend naturally to multi-agent scenarios involving more than two vehicles. Specifically, inspired by the communication paradigm in [1], we construct a fully connected communication graph in which each agent broadcasts its request messages (i.e., QSM and SCM) to all others. The necessity of communication between the *i*-th and *j*-th agents is determined by the overlap between the information possessed by agent *i* and the information required by agent *j*, which is quantified by the identified non-redundant tokens.
>
> ---
>
> **Q3 Ablation studies.**
>
> We thank the reviewer for this suggestion. We agree that stronger ablations can clarify component contributions, especially for grouping and importance estimation.
>
> | Variant / Setting        | L2 (m) ↓ | CR (%) ↓ |
> |-------------------------|----------|----------|
> | Dense sharing baseline  | 4.99     | 3.00     |
> | Full V2V-CCM            | 5.21     | 6.28     |
> | QSM-only pruning        | 6.11     | 7.67     |
> | SCM-only pruning        | 6.23     | 7.89     |
> | QSM + SCM (no grouping)             | 5.88     | 7.13     |
>
> Removing object-scene grouping leads to clear performance degradation, highlighting the importance of group-level importance estimation. Moreover, instance-level scoring may incorrectly prune complementary multi-view information. To validate this, we conduct an additional experiment using instance-level tokens (object tokens) as indicators, resulting in L2 = 7.81 m and CR = 8.25%, which is worse than the SCM-only setting. This confirms that scene-level grouping is more effective. We will include more comprehensive ablations in the camera-ready version.
>
> ---
>
> **References**
>
> [1] Where2comm: Communication-efficient collaborative perception via spatial confidence maps, NeurIPS 2022.

---

> > ### Author Rebuttal · Reviewer_BZpB · 2026-04-01
> >
> > Thank you for your reply. The experimental results address my concerns. I have carefully read the authors' response as well as the discussion with other reviewers. I have decided to raise my score to relatively positive.

---

> > > ### Author Response · Authors · 2026-04-02
> > >
> > > We sincerely thank you for considering our rebuttal and for the increased score. We greatly appreciate your feedback and the time you have devoted to evaluating and helping improve our work.

---

### Official Review · Reviewer_B4ZC · 2026-03-13

**Soundness:** 3
**Presentation:** 2
**Significance:** 3
**Originality:** 3
**Overall Recommendation:** 4
**Confidence:** 5

**Summary:**

The paper proposes V2V-CCM for cooperative autonomous driving with multimodal LLMs, aiming to reduce inefficient dense token sharing between vehicles. It introduces two compact messages: QSM for question-relevant semantics and SCM for spatial coverage to avoid redundant communication. These messages guide message-conditioned token pruning, which selects the most useful LiDAR-derived tokens under a communication budget. Experiments on V2V-QA and V2V-GoT-QA show better performance retention than prior pruning baselines across multiple pruning ratios. Overall, the work presents a communication-aware token pruning strategy for multi-agent LiDAR-language cooperation in autonomous driving.

**Compliance With Llm Reviewing Policy:**

Affirmed.

**Key Questions For Authors:**

1-Can you report direct efficiency metrics, not only task performance under pruning? Specifically, please include communication payload per agent, end-to-end latency, and ideally FLOPs or wall-clock inference time before and after V2V-CCM. It would also be helpful to separate the overhead of generating QSM/SCM from the actual savings achieved through token pruning.

2-Can you clarify the contribution of each component through stronger ablations? In particular, please isolate QSM-only pruning, SCM-only pruning, and the effect of removing object-scene grouping, and also report sensitivity to both λ and γ. This would make it clearer which parts of the method drive the overall gains

3-How do you position V2V-CCM relative to stronger communication-aware baselines specifically designed for cooperative perception, rather than mostly generic token-pruning methods? The current comparisons are useful, but several baselines appear to be generic token pruning approaches adapted to this setting. Please clarify whether there are more directly relevant communication-selection or bandwidth-aware cooperative perception/planning baselines, and if not, explain why these are the most appropriate comparisons.

**Limitations:**

No. The paper does mention one technical limitation, that it reduces communication-stage cost but does not explicitly optimize LLM inference-stage cost.
Since the method deliberately prunes shared information, the paper should discuss the risk that safety-critical but low-scoring tokens are dropped, especially in rare, occluded, or adversarial scenarios.
The authors should acknowledge that evaluation is limited to two cooperative-driving benchmarks and may not reflect deployment across different traffic densities, sensing stacks, or agent counts.
A stronger discussion would note possible downstream harms from system failure, including collision risk, unequal performance across environments, and over-reliance on communication-enabled autonomy in settings where infrastructure is unreliable.

**Strengths And Weaknesses:**

Soundness:
The proposed remedy is technically coherent: agents exchange two compact messages, a question semantic message and a spatial coverage message, then prune tokens by combining question relevance and spatial redundancy at both group and token levels under a global budget. The formulation is simple and plausible, and the method is evaluated on two relevant cooperative-driving benchmarks with multiple pruning ratios and several baselines. Empirically, the results are reasonably strong: across 30%, 50%, and 70% pruning, V2V-CCM is consistently better than Random, ToMe, VisionZip, and DivPruner, and at light pruning it often stays close to the unpruned baseline, especially on planning metrics. The λ ablation is also useful because it supports the claim that balancing relevance and redundancy matters.
The paper repeatedly claims reduced communication and inference cost, but the experiments mainly report downstream task metrics, not explicit latency, FLOPs, or end-to-end runtime. also, would have liked stronger ablations isolating QSM alone, SCM alone, object-scene grouping alone, and the effect of γ in addition to λ.

Presentation
The overall story is easy to follow, Figures 1 and 2 help communicate the design clearly, and the optimization objective is simple enough to understand without excessive notation.
More substantively, the paper would benefit from clearer reporting of implementation details: exact token counts before and after pruning, message sizes, how group neighborhoods are defined in practice, how the top-K solution is implemented across agents, and whether retraining/fine-tuning is required.
Significance
This is an important problem. If V2V MLLMs are to be useful in realistic cooperative driving, communication efficiency is not optional, and the paper addresses that bottleneck directly. The strongest aspect of the significance is that it brings together communication-aware design and language-conditioned reasoning in a setting where both matters.
 Because the paper does not directly quantify actual runtime, bandwidth, or deployment-level savings, it stops short of proving that the approach materially changes the operating point of a real system. Also, the evaluation is tied to two benchmarks derived from the same general ecosystem of V2V LiDAR QA/planning tasks, so the breadth of impact remains somewhat narrow for now.
Originality
The work is meaningfully original at the level of problem framing and method combination. That shift from dense token exchange to request-conditioned selective exchange is a good fit for cooperative driving, and combining question semantics with spatial coverage for pruning LiDAR-derived multi-agent tokens is a sensible and nontrivial extension beyond standard single-stream VLM pruning.
The novelty lies in how they are assembled for V2V MLLM communication, not in a fundamentally new learning principle. That is still a legitimate contribution, but I would view it as a well-motivated systems/method integration paper rather than a highly novel conceptual leap.

---

> ### Author Rebuttal · Authors · 2026-03-30
>
> We appreciate the professional and insightful comments. We address each comment as follows:
>
> ---
>
> **Q1 Efficiency metrics and analysis**
>
> We thank the reviewer for this suggestion. We agree that reporting direct efficiency metrics is important for a comprehensive evaluation. We provide representative results to illustrate the impact of token pruning on communication and computation efficiency, including communication bandwidth (Comm. BW), TFLOPs, TTFT (Time to First Token), and speedup:
>
> | Model    | Pruning Ratio | Comm. BW (MB) ↓ | TFLOPs ↓ | TTFT (ms) ↓ | Speedup ↑ |
> |----------|---------------|-----------------|----------|-------------|-----------|
> | V2V-LLM  | No Pruning    | 0.4068          | 15.12   | 45.5        | 1.00×     |
> |          | 30%           | 0.285           | 10.6    | 36.0        | 1.26×     |
> |          | 50%           | 0.2034          | 7.99    | 30.2        | 1.51×     |
> |          | 70%           | 0.122           | 4.5     | 22.0        | 2.07×     |
> | V2V-GoT  | No Pruning    | 0.4068          | 16.04   | 48.3        | 1.00×     |
> |          | 30%           | 0.285           | 11.2    | 38.0        | 1.27×     |
> |          | 50%           | 0.2034          | 8.90    | 31.9        | 1.51×     |
> |          | 70%           | 0.122           | 5.0     | 23.0        | 2.10×     |
>
> These results show that increasing the pruning ratio consistently reduces communication payload, computational cost, and prefilling latency. The overhead of QSM/SCM generation is relatively small compared to the savings from token pruning. We will provide a more detailed breakdown of this overhead in the camera-ready version.
>
> ---
>
> **Q2  Component-wise ablation and contribution analysis**
>
> We thank the reviewer for this helpful suggestion. We agree that additional ablation studies can better clarify the contribution of each component. The effect of λ is partially discussed in Table 2 of the main paper. Here, we summarize the relevant settings under the same configuration (70% pruning ratio on the V2V-QA dataset, Q5 Planning):
>
> | Variant / Setting                     | L2 (m) ↓ | CR (%) ↓ |
> |--------------------------------------|----------|----------|
> | Dense sharing baseline               | 4.99     | 3.00     |
> | Full V2V-CCM     | 5.21     | 6.28     |
> | QSM-only pruning   | 6.11     | 7.67     |
> | SCM-only pruning                     | 6.23     | 7.89     |
> | QSM + SCM (no grouping)              | 5.88     | 7.13     |
> | λ = 0.0, γ = 1.0                     | 6.08     | 7.54     |
> | λ = 1.0, γ = 0.0                     | 5.91     | 7.33     |
> | λ = 1.0, γ = 0.5                     | 5.57     | 6.51     |
> | λ = 0.5, γ = 1.0                     | 5.61     | 6.81     |
>
> The results indicate that both QSM and SCM contribute to performance improvements, while neither alone is sufficient. Removing object-scene grouping also degrades performance, highlighting the importance of grouping. Furthermore, balancing λ and γ is critical—moderate values (e.g., λ = 1.0, γ = 0.5) achieve a better trade-off between planning performance and communication efficiency.
>
> We will include a more comprehensive ablation study with clearer component isolation and additional settings in the camera-ready version.
>
> ---
>
> **Q3 Positioning relative to communication-aware V2V baselines**
>
> We thank the reviewer for this important suggestion. We agree that including cooperative perception baselines could further strengthen the evaluation. However, most existing V2V methods are designed for 3D object detection and are tightly coupled with specific detector architectures. For example, How2Comm [1] introduces attention mechanisms within 3D detection pipelines.
>
> While some works (e.g., Where2Comm [2]) aim to reduce communication bandwidth, they typically operate at the feature level (e.g., BEV or voxel features), which is substantially larger than token-level representations and does not reduce inference cost in MLLM-based reasoning. In contrast, our work focuses on MLLM-based cooperative reasoning and introduces a token-level communication paradigm that reduces both communication bandwidth and inference cost.
>
> Therefore, we adopt token-pruning-based methods as the most appropriate baselines, as they enable direct comparison in terms of token efficiency and inference acceleration within the same modeling paradigm.
>
> ---
>
> **References**
> [1] How2Comm: Communication-Efficient and Collaboration-Pragmatic Multi-Agent Perception. NeurIPS 2023.
> [2] Where2Comm: Communication-Efficient Collaborative Perception via Spatial Confidence Maps. NeurIPS 2022.

---

> > ### Author Rebuttal · Reviewer_B4ZC · 2026-04-08
> >
> > Thank you for the clear rebuttal. My main concerns were about (1) reporting direct efficiency metrics, (2) providing stronger component-wise ablations, and (3) clarifying the choice of baselines. The rebuttal addresses these points with additional quantitative results. The rebuttal addressed these points with additional quantitative results and helpful clarification. Overall, it resolved my main concerns, and I appreciate the authors’ effort to strengthen the paper.

---

### Official Review · Reviewer_Q3ns · 2026-03-15

**Soundness:** 3
**Presentation:** 3
**Significance:** 3
**Originality:** 3
**Overall Recommendation:** 4
**Confidence:** 4

**Summary:**

The paper introduces V2V-CCM, a communication-efficient framework for vehicle-to-vehicle (V2V) cooperative autonomous driving using Multimodal Large Language Models (MLLMs). While MLLMs offer powerful reasoning for joint perception and planning, they typically suffer from high communication overhead due to dense token sharing. To address this, the authors propose a dual-stage communication paradigm where agents first exchange compact "request messages" to identify and prune redundant visual tokens before full transmission.

Specifically, the framework utilizes two types of messages:
- Question Semantic Message (QSM): Encodes the global intent of the natural language question to prioritize task-relevant tokens.
- Spatial Coverage Message (SCM): Tokenizes LiDAR bird's-eye-view (BEV) features to identify regions already observed by other agents, thus avoiding redundant data transmission.

These messages guide a Message-Conditioned Token Pruning (MCTP) strategy that optimizes token selection under a global budget. Experiments on V2V-QA and V2V-GoT-QA datasets demonstrate that V2V-CCM maintains high performance in perception and planning tasks while significantly reducing communication and inference costs compared to existing pruning methods.

**Compliance With Llm Reviewing Policy:**

Affirmed.

**Key Questions For Authors:**

- Vision-Based Adaptability: Could the SCM (Spatial Coverage Message) be effectively adapted for vision-based (camera) inputs, where depth estimation is more uncertain compared to LiDAR's precise geometric data?
- End-to-End Synergy: How would the token pruning strategy perform if integrated into a fully end-to-end differentiable driving model, rather than relying on a pre-trained 3D object detector?
- Latency Trade-offs: Can you provide a detailed breakdown of the end-to-end latency (in milliseconds) comparing the "dense sharing" baseline vs. the "dual-stage" CCM, including the time taken for the initial request message exchange?

**Limitations:**

- Modality Constraint: The current evaluation is restricted to LiDAR data, which may not reflect the constraints of camera-only V2V systems.
- Dataset Specificity: The results are reported primarily on the V2V-split of specific datasets (V2V-QA, V2V-GoT-QA). The robustness of the pruning strategy in more diverse or unmapped environments remains to be fully explored.
- Static Budgeting: The "Global Budget" for tokens appears to be a hyperparameter. In highly dynamic environments, a fixed budget might be suboptimal; a more adaptive budget based on scene complexity would be an ideal but currently unaddressed extension.

**Strengths And Weaknesses:**

**Strengths**
- Soundness: The proposed dual-stage communication strategy is technically sound, leveraging both semantic (question-based) and geometric (LiDAR-based) constraints to prune tokens effectively. The use of both QSM and SCM provides a holistic approach to redundancy.
- Significance: Reducing communication bandwidth is a critical challenge for real-world V2V deployment. This paper provides a practical solution that allows MLLMs to be more scalable in multi-agent environments.
- Originality: Integrating a dual-stage "request-response" communication flow specifically for MLLM token pruning in a V2V context is a novel contribution that moves beyond standard feature-sharing or general-purpose token pruning.

**Weaknesses**
- Input Modality Choice: The framework relies on LiDAR-based input (via PointPillars). Given the recent industry trend and research shift towards vision-centric (camera-only) or multi-modal systems, the reliance on LiDAR might limit the generalizability of the findings to more cost-effective vision-based autonomous systems.
- Lack of Full End-to-End Integration: While the paper addresses planning (Q5 and Q9), modern autonomous driving research is heavily moving toward unified end-to-end frameworks where perception and control are more tightly coupled. The current framework feels somewhat modular (detector $\to$ projector $\to$ MLLM), which may not fully capture the benefits or challenges of a truly end-to-end system.Complexity of Dual-Stage Flow: While the dual-stage process saves bandwidth, it introduces additional latency due to the "request" round-trip. The paper could benefit from a more detailed analysis of the trade-off between communication bits saved versus the time delay introduced by the two-step handshake.

---

> ### Author Rebuttal · Authors · 2026-03-30
>
> We appreciate the professional and insightful comments. We address each comment as follows:
>
> **W1 & Q1 Generalization to camera-based modalities**
>
> We agree that incorporating camera modalities could further improve the generalizability of our method. Currently, existing LLM-based V2V datasets primarily provide LiDAR modalities. To demonstrate the effectiveness of our approach, we evaluate QSM in a single-agent setting using the DriveLM [1] dataset (note that SCM requires multi-agent scenarios to identify redundancy) with the EM-VLM4AD [2] model. Representative results are shown below:
>
> | Method                  | BLEU-4 ↑ | ADE ↓ |
> |-------------------------|----------|-------|
> | EM-VLM4AD (Baseline)    | 45.36    | 1.85  |
> | + QSM (Ours)            | 44.81    | 2.07  |
>
> These results suggest that components of our method can generalize to camera-based modalities.
>
> ---
>
> **W2 & Q2 Integration with end-to-end driving frameworks**
>
> We thank the reviewer for the suggestion to consider fully end-to-end differentiable driving models, which could further improve the integration of our framework. QSM is expected to remain effective in identifying redundant tokens, as discussed in W1 & Q1. For SCM, although explicit features may not always be required, intermediate representations can still serve as effective indicators, since feature extraction from environmental observations is fundamental in most pipelines.
>
> For example, in BEV-based LLMs, where camera and LiDAR features are projected into a shared BEV space [2], SCM can be applied by selecting appropriate intermediate features. Similarly, in frameworks such as UniAD [3], perception features can also serve as indicators. For sparse scene representations (e.g., SparseDrive [4]), further investigation is needed to adapt the design of scene-level indicators.  We consider this an important direction for future work and will explore it further in subsequent studies.
>
> ---
>
> **W3 & Q3 Latency comparison between dual-stage and dense sharing**
>
> We agree that a practical pipeline comparison between the two frameworks is important. We compare the “dense sharing” baseline and our dual-stage V2V-CCM from the following perspectives:
>
> **(1) Communication bandwidth:**
> For a single agent pair, communication is reduced from 0.4068 MB to 0.122 MB (at 70% pruning). As the number of agents increases, the cost grows quadratically due to all-to-all communication. For N agents, the total communication cost is reduced from 0.4068 × (N−1)² to 0.122 × (N−1)², significantly reducing bandwidth and redundancy.
>
> **(2) End-to-end latency:**
> We evaluate the full pipeline on both edge and server devices. The dual-stage design introduces an additional ~15 ms latency for request exchange in standard network settings. However, it substantially reduces inference time through token pruning. We use TTFT (Time to First Token) as the evaluation metric:
>
> - **H200:**
>   V2V-LLM: 45.5 ms → 22.0 ms (−23.5 ms)
>   V2V-GoT: 48.3 ms → 23.0 ms (−25.3 ms)
>
> - **Jetson AGX Orin:**
>   V2V-LLM: 5.33 s → 4.83 s (−0.50 s)
>   V2V-GoT: 6.14 s → 5.55 s (−0.59 s)
>
> We acknowledge that there is a trade-off between communication overhead (due to the additional stage, which may increase under poor network conditions) and device-dependent inference time (ranging from milliseconds to seconds). However, overall, our method reduces bandwidth while maintaining—and in many cases reducing—total latency.
>
> **(3) Scalability and system efficiency:**
> Beyond bandwidth and latency reduction, the proposed “request-then-share” paradigm improves system efficiency. As the number of agents increases, broadcasting full tokens leads to quadratic growth in communication cost and substantial redundancy. Moreover, in practical implementations, directly sharing tokens without prior coordination may introduce redundancy and inefficiency. In contrast, our approach enables selective communication based on demand, making it more scalable and suitable for V2V scenarios.
>
> ---
>
> **References**
> [1] DriveLM: Driving with Graph Visual Question Answering, ECCV 2024.
> [2] Multi-frame, Lightweight & Efficient Vision-Language Models for Question Answering in Autonomous Driving, 2024.
> [3] Planning-Oriented Autonomous Driving, CVPR 2023.
> [4] SparseDrive: End-to-End Autonomous Driving via Sparse Scene Representation, ICRA 2025.

---

> > ### Author Rebuttal · Reviewer_Q3ns · 2026-04-04
> >
> > I thank the authors for their detailed rebuttal. The additional experiments on camera-based modalities and the specific latency breakdown effectively address my concerns regarding generalizability and real-time trade-offs. The clarification on end-to-end integration is also reasonable. I am satisfied with the responses and will maintain my score as Weak Accept.

---

> > > ### Author Response · Authors · 2026-04-05
> > >
> > > We sincerely thank you for considering our rebuttal and for maintaining your score, and we greatly appreciate your valuable feedback and time in helping improve our work.

---

### Decision · Program_Chairs · 2026-04-30

**Decision:**

Accept (regular)

**Comment:**

The paper proposes V2V-CCM, a novel communication-conditioned token pruning framework for Multimodal Large Language Models (MLLMs) in vehicle-to-vehicle cooperative driving. Initially, the reviewing committee recognized the technical soundness and originality of the dual-stage message design (QSM and SCM), but raised valid questions regarding the lack of direct efficiency metrics (such as latency, bandwidth, and FLOPs), missing component-wise ablations, and the framework's adaptability to camera-only or end-to-end pipelines. During the rebuttal phase, the authors provided a highly effective response, supplying detailed quantitative tables demonstrating significant reductions in Time to First Token (TTFT) and communication payload, alongside comprehensive ablations isolating the effects of object-scene grouping and balancing hyperparameters like $\lambda$ and $\gamma$. The authors also successfully clarified how the method preserves complementary multi-view features and demonstrated its potential to generalize to camera modalities. Following the rebuttal, all reviewers actively engaged with the new evidence, confirmed that their core concerns were adequately resolved, and unanimously finalized their scores at Weak Accept. Aligning with this positive consensus and the authors' diligent revisions, the paper is recommended for **weak acceptance** as a solid and practical contribution to communication-efficient cooperative perception.